# Clinical Application of Whole Exome Sequencing to Identify Rare but Remediable Neurologic Disorders

**DOI:** 10.3390/jcm9113724

**Published:** 2020-11-20

**Authors:** Min-Jee Kim, Mi-Sun Yum, Go Hun Seo, Yena Lee, Han Na Jang, Tae-Sung Ko, Beom Hee Lee

**Affiliations:** 1Department of Pediatrics, Asan Medical Center Children’s Hospital, Ulsan University College of Medicine 88, Olympic-ro 43-Gil, Songpa-Gu, Seoul 05505, Korea; pradoxwh@naver.com (M.-J.K.); janghannah85@gmail.com (H.N.J.); tsko@amc.seoul.kr (T.-S.K.); 23billion Inc., Seoul 06193, Korea; ghseo@3billion.io (G.H.S.); mdlbh@hanmail.net (B.H.L.); 3Department of Genetics, Asan Medical Center, Ulsan University College of Medicine, 88, Olympic-ro 43-Gil, Songpa-Gu, Seoul 05505, Korea; refreshyn@gmail.com

**Keywords:** whole exome sequencing, rare genetic diseases, neurogenetic disorders, congenital myasthenic syndrome, autosomal recessive dopa-responsive dystonia, epileptic encephalopathy 47, episodic ataxia type II

## Abstract

Background: The aim of this study was to describe the application of whole exome sequencing (WES) in the accurate genetic diagnosis and personalized treatment of extremely rare neurogenetic disorders. Methods: From 2017 to 2019, children with neurodevelopmental symptoms were evaluated using WES in the pediatric neurology clinic and medical genetics center. The clinical presentation, laboratory findings including the genetic results from WES, and diagnosis-based treatment and outcomes of the four patients are discussed. Results: A total of 376 children with neurodevelopmental symptom were evaluated by WES, and four patients (1.1%) were diagnosed with treatable neurologic disorders. Patient 1 (Pt 1) showed global muscle hypotonia, dysmorphic facial features, and multiple anomalies beginning in the perinatal period. Pt 1 was diagnosed with congenital myasthenic syndrome 22 of PREPL deficiency. Pt 2 presented with hypotonia and developmental arrest and was diagnosed with autosomal recessive dopa-responsive dystonia due to TH deficiency. Pt 3, who suffered from intractable epilepsy and progressive cognitive decline, was diagnosed with epileptic encephalopathy 47 with a heterozygous *FGF12* mutation. Pt 4 presented with motor delay and episodic ataxia and was diagnosed with episodic ataxia type II (heterozygous *CACNA1A* mutation). The patients’ major neurologic symptoms were remarkably relieved with pyridostigmine (Pt 1), levodopa (Pt 2), sodium channel blocker (Pt 3), and acetazolamide (Pt 4), and most patients regained developmental milestones in the follow-up period (0.4 to 3 years). Conclusions: The early application of WES helps in the identification of extremely rare genetic diseases, for which effective treatment modalities exist. Ultimately, WES resulted in optimal clinical outcomes of affected patients.

## 1. Introduction

Since the advent of massively parallel, genomic-sequencing techniques in the early 2000s, the diagnostic process and speed at which genetic diseases are identified have greatly improved. In particular, either whole exome sequencing (WES) or whole genome sequencing (WGS) can reveal more than 5000 phenotypically and genetically diverse conditions with a single test. The diagnostic accuracy of WES and WGS is approximately 35–40% [1,2,3,4,5]. Genome sequencing enhances the ability to identify previously unrecognized, rare genetic disorders [6,7]. Approximately 6000 to 8000 rare diseases have been discovered and 80% of them have identifiable genetic backgrounds [8,9]. Among the 6700 phenotypes listed in the Online Mendelian inheritance in Man (OMIM; www.omim.org), about 40% are associated with neurologic manifestations, which are the most common clinical features of rare diseases.

Identification of neurological disorders by genomic sequencing has helped clinicians understand the underlying pathophysiologies, resulting in personalized treatment for some of these rare diseases. Although the proportion of neurogenetic disorders that are treatable is very small, identification of these extremely rare genetic conditions can dramatically change the quality of life in affected patients. Here, we present four patients who were diagnosed with extremely rare but treatable neurologic disorders identified by WES. This study demonstrates the clinical benefit of WES in pediatric patients with neurological manifestations.

## 2. Materials & Methods

A total of 376 children with neurodevelopmental symptoms, including delayed development or hypotonia, were evaluated using WES (patient 1, 2, 3) and panel test (patient 4) in the pediatric neurology clinic and medical genetics center at Asan Medical Center Children’s Hospital, Seoul, Korea, from 2017 to 2019.

For WES, genomic DNA was isolated from either whole blood or saliva. Majority of exons in all human genes (approximately 22,000) were captured using a SureSelect kit (Version C2; Agilent Technologies, Inc., Santa Clara, CA, USA) and sequenced using a NovaSeq platform (Illumina, San Diego, CA, USA). Raw genome sequences were aligned to a reference sequence (NCBI genome assembly GRCh37; accessed in February 2009). For panel test, genomic DNA extracted from peripheral blood leukocytes. Approximately 62,000 exomes were captured using the TruSight One Panel (Illumina Inc., San Diego, CA, USA), which enriches a 12-Mb region spanning 4813 genes (gene lists in Appendix A and the read depth and coverage in Appendix A).

The streamlined, automated variant prioritization software system, termed EVIDENCE (3bilion Inc., Seoul, Korea) [10] used to identify the candidate variants is represented in Figure 1. The software program analyses over 100,000 variants, according to ACMG guidelines, prioritizes the variants based on each phenotype of each patient and interprets these variants accurately and consistently. Each variant identified by WES and panel test (only patients 4), the filtering process was confirmed using Sanger sequencing of samples from patients and families. This study was approved by the Institutional Review Board for Human Research of Asan Medical Center (IRB number 2017-0988).

## 3. Results

Among the 376 children with neurological symptoms who underwent WES, the diagnostic yield of WES for these cohort were 42.7% [10] and four patients (1.1%) were diagnosed with actionable neurologic disorders with precision treatment (Table 1).

### 3.1. Clinical Presentation of the Four Patients

#### 3.1.1. Patient 1. Myasthenic Syndrome, Congenital, 22 (OMIM 616224)

Patient 1 was male and the first child of nonconsanguineous Korean parents (Table 1). He was born at 34 gestational weeks with a birth weight of 1620 g (Standard deviation (SD), −4.39), a height of 41 cm (SD, −4.69), and a head circumference of 30.5 cm (SD, −3.12). Cesarean section delivery was performed because of oligohydramnios. Immediately after birth, he was admitted to the neonatal intensive care unit due to prematurity and respiratory distress. On physical examination at birth, dysmorphic features were noted, including thick arched eyebrows, hypertelorism, a broad prominent nasal bridge, low set ears, a cleft and high arched palate, and a short webbed neck. The patient showed global muscle hypotonia with limited spontaneous movement and decreased deep tendon reflex. The patient also had multiple additional anomalies, including hearing disturbance, supravalvular aortic stenosis, a tethered spinal cord, cryptorchidism, and duplex kidney. The brain magnetic resonance image (MRI) at 1 month of age showed no specific abnormality. Laboratory tests, including a metabolic profile, plasma amino acids, urine organic acids, plasma acylcarnitines, and thyroid function were unremarkable. The karyotyping, multiplex ligation-dependent probe amplification (MLPA) analysis for 26 microdeletion syndromes, and the diagnostic exome sequencing of 4813 OMIM genes were also normal. At 2 years old, the patient presented with global developmental delay, barely sat with support, and only spoke a single word.

#### 3.1.2. Patient 2. Autosomal Recessive Dopa-Responsive Dystonia (OMIM 605407)

Patient 2 was female and the first child of healthy nonconsanguineous Korean parents (Table 1). She was born at 39 gestational weeks with a birth weight of 2400 g (SD, −0.75). Cesarean section delivery was performed due to oligohydramnios. The patient’s developmental milestones were delayed; the patient did not have head control and could not roll over at 9 months. On physical examination, she showed decreased muscle tone with lagging head during traction of both arms and normal deep tendon reflex. There was no spasticity or dystonic movement. Genetic testing, including karyotyping and mitochondrial DNA sequencing, was normal.

#### 3.1.3. Patient 3. Epileptic Encephalopathy 47 (OMIM 617166)

Patient 3 was female and the first child of healthy nonconsanguineous Korean parents (Table 1). She was born at 39 gestational weeks with a birth weight of 3200 g (SD, −0.06), a height of 44 cm (SD, −2.76), and a head circumference 34 cm (SD, 0.1). Pregnancy, labor, and vaginal delivery were uneventful. The clonic movement of the left arm with left eyeball deviation developed beginning at the age of 3 days. The brain MRI was unremarkable. The electroencephalogram (EEG) revealed an encephalopathic pattern with intermittent suppression-burst background and occasional sharp wave discharges from the left occipital and central head areas (Figure 2). Laboratory tests, including metabolic profiles, thyroid function, and karyotyping were unremarkable. At 2 months, the patient’s seizures evolved to the generalized tonic-clonic type and were refractory to multiple antiepileptic drugs (AEDs), including phenobarbital, levetiracetam, clonazepam, and clobazam. Phenytoin introduction during acute repetitive seizure events at 2 years of age controlled the patient’s seizures, but the seizures recurred depending on the dose of phenytoin. The patient had severe global developmental delay; she could walk alone at age 2.6 years, she could climb stairs but could not go down, and she spoke a meaningless babbling at 8 years.

#### 3.1.4. Patient 4. Episodic Ataxia, Type II (OMIM 108500).

Patient 4 was male and the third child of healthy nonconsanguineous Korean parents (Table 1). His birth was uneventful at 37 gestational weeks with a birth weight of 3380 g (SD, 0.07). Beginning at 12 months, the patient had ataxic attacks with dysarthria and medial deviation of both eyes four times a week. The patient could barely walk alone at age 17 months. Laboratory tests, including a brain MRI and MR angiography, electroencephalography, electrocardiogram, complete metabolic profile, and thyroid function, were unremarkable (Figure 3). The karyotyping and MLPA for 26 microdeletion syndromes were also normal.

### 3.2. Genetic Diagnosis

The whole exomes carried an average of 82,384 variants per patient. Variants with a frequency of 5% or higher in the minor allele frequency were excluded, eliminating nearly 98% of the variants. Approximately 8174 variants remained for each patient. After genes were matched with known diseases, about 2202 variants remained. Finally, average 58 disease-variant pairs were curated manually, after excluding variants with low impact, including likely benign, benign, and non-coding variants with low evidence according to the ACMG guidelines, and filtering by known inheritance pattern. Among these 58 disease-variant pairs, candidate genetic variants were selected based on the relationship between the genes and patient phenotypes. The numbers of variants after each filtering process are presented in Table 2.

#### 3.2.1. Patient 1. Myasthenic Syndrome, Congenital, 22 (OMIM 616224)

When Patient 1 was 2.6 years old, WES revealed a homozygous variant in the *PREPL* gene (NM_001171603.1) on chromosome 2, with a c.1940G>A (p.Arg647Gln) mutation inherited from both parent carriers (Figure 4). The prolyl endopeptidase-like (PREPL) protein is a cytoplasmic serine hydrolase structurally belonging to the oligopeptidase family. The c.1940G>A variant was predicted to alter the protein function by in silico analysis (MutationTaster; Polyphen-2 and SIFT) and has not been reported in the general population cohort (https://gnomad.broadinstitute.org/). Thus, this variant was classified as “uncertain significance” according to the ACMG guidelines [11]. Although this variant remains uncertain significance, this variant might be highly likely relevant to the patient’s phenotype, since trans configuration for the identified variant was confirmed and the patient also showed highly specific symptoms related to this disease. Therefore, he was diagnosed with congenital myasthenic syndrome 22 (CMS22).

#### 3.2.2. Patient 2. Autosomal Recessive Dopa-Responsive Dystonia (OMIM 605407)

When Patient 2 was 10 months old, WES revealed compound heterozygous variants in *TH* (NM_199292.2) on chromosome 11, c.1393T>C variant (p.Ser465Pro) inherited from the father and a c.698G>A (p.Arg233His) variant inherited from the mother (Figure 4). Tyrosine hydroxylase converts L-tyrosine to L-3,4-diyhdroxyphenylalanine (L-DOPA), an essential step in the formation of dopamine and other catecholamines. The c.1393T>C and c.698G>A variants are predicted to alter the protein function based on pathogenicity predictions (REVEL score 0.846 and 0.955, respectively). The c.1393T>C variant has not been previously reported in medical literatures. On the other hand, the c.698G>A has been reported to be homozygous or compound heterozygous variant in many affected individuals [12]. These two variants showed an extremely low frequency in the general population cohort (https://gnomad.broadinstitute.org/). Thus, c.1393T>C and c.698G>A variants were classified as “likely pathogenic” and “pathogenic,” respectively, according to the ACMG guidelines [11]. Therefore, the patient was finally diagnosed with autosomal recessive dopa-responsive dystonia (AR-DRD).

#### 3.2.3. Patient 3. Epileptic Encephalopathy 47 (OMIM 617166)

When Patient 3 was 8 years old, WES revealed a heterozygous missense variant in *FGF12* (NM_021032.4) on chromosome 3, c.341G>A (p.Arg114His) mutation in the A-isoform, and this variant was not present in the unaffected parents (Figure 4). *FGF12* encoded for a member of the fibroblast growth factor homologous factor (FHF) family, which interacts with the C-terminal tails of voltage-gated sodium channels and modulates fast inactivation [13,14]. The c.341G>A variant was predicted to damage protein function (REVEL score 0.548) [14,15] and was previously reported as a pathogenic variant [14,16]. In addition, this variant has not been reported in the general population cohort (https://gnomad.broadinstitute.org/). This variant was classified as “pathogenic” according to the ACMG guidelines [11]. Therefore, the patient was finally diagnosed with early infantile epileptic encephalopathy-47 (EIEE-47).

#### 3.2.4. Patient 4. Episodic Ataxia, Type II (OMIM 108500)

When Patient 4 was 4 years old, panel test for 62,000 exomes in 4813 genes revealed a heterozygous missense variant in *CACNA1A* (NM_001127221.1) on chromosome 19, c.3790G>A (p.Glu1264Lys) mutation. This variant was not found in the unaffected parents (Figure 4). *CACNA1A* encodes for the transmembrane pore-forming α1A subunit of the P/Q type or CaV2.1 voltage-gated calcium channel (VGCC) [17]. These channels are expressed in a large variety of neurons and play an important role in membrane excitability, neurotransmitter release, and gene expression [18]. The c.3790G>A mutation was predicted to damage protein function (REVEL score 0.922). In addition, this variant has not been reported in the general population cohort (https://gnomad.broadinstitute.org/). Thus, this variant was classified as “likely pathogenic” according to the ACMG guidelines [11]. Patient 4 was finally diagnosed with episodic ataxia, Type II (EA 2).

### 3.3. Post-Diagnosis Treatment and Clinical Outcome

#### 3.3.1. Patient 1. Myasthenic Syndrome, Congenital, 22 (OMIM 616224)

After the above diagnosis, Patient 1 received pyridostigmine (7.5 mg/kg/day) beginning at age 2.6 years. Six months after treatment, the patient could sit alone, crawl, and lift his arms to shoulder level.

#### 3.3.2. Patient 2. Autosomal Recessive Dopa-Responsive Dystonia (OMIM 605407)

A therapeutic trial of levodopa (combined with a decarboxylase inhibitor, initial dose 1 mg/kg/day with slow titration) was started at 10 months of age in Patient 2. The patient began to control her head for a longer period and rolled over after 1 month of treatment. At the age of 14 months, the patient could sit alone and stand with support. The patient showed nearly normal muscle tone and could speak several words when treated with levodopa (8 mg/kg/day) and a small dose of MAO inhibitor (0.378 mg/kg/day).

#### 3.3.3. Patient 3. Epileptic Encephalopathy 47 (OMIM 617166)

After diagnosis, the patient was maintained on phenytoin, which is appropriate for treating EIEE-47. At the most recent evaluation at 9 years old, the patient’s height was 131.4 cm (SD, −0.04), body weight was 29.5 kg (SD, −0.01), and head circumference was 50 cm (SD, −0.6). The patient still exhibited a wide-based, unstable gait but could jump, take off her clothes without help, and speak several words.

#### 3.3.4. Patient 4. Episodic Ataxia, Type II (OMIM 108500)

Acetazolamide (16 mg/kg/day) was administrated to Patient 4 immediately after diagnosis and his ataxic episodes developed only after strenuous exercise. The patient slowly improved in his motor milestones.

## 4. Discussion

In the current study, we describe four patients with extremely rare but treatable neurological disorders, which were identified using WES. After the genetic confirmation, prompt medical intervention and treatment improved the devastating disease course of each patient. Our experience represents additional examples of patient-tailored therapies, which have created a major paradigm shift in medicine [19]. Among the neurogenetic diseases, glucose transporter type-1 deficiency, X-linked adrenoleukodystrophy, and dopa-responsive dystonia are considered successful models of these patient-tailored therapies [20,21]. The number of neurogenetic diseases is increasing with the development of new therapeutic modalities [22,23]. In our study cohort, these patient-tailored therapies have remarkably changed the clinical course in 4 patients (1.1%) among the 376 children with neurodevelopmental symptoms that were evaluated using WES.

WES facilitates precision medicine and has changed screening and management in 41–44% of pediatric neurologic diseases, including early-onset epilepsy and neurometabolic diseases [20,21,24]. Still, as in our patient cohort, only a small proportion of patients have the opportunity to receive effective or curative medical treatment. Treatment targeting the identified abnormality at the cellular or molecular level is available for only 7–14% of patients [20,24]. However, the number of patients with precision medicine decreases even further for extremely rare and remediable diseases.

CMS, the diagnosis for Patient 1, belongs to a genetically and phenotypically heterogeneous group of early-onset neuromuscular transmission diseases [25,26]. Currently, more than 30 genes are responsible for the different subtypes of CMS and these CMS subtypes share the clinical features of muscle weakness and electrophysiological abnormalities [27]. CMS22 is an autosomal recessive ultra-rare disorder caused by an isolated PREPL deficiency. The PREPL deficiency is characterized by severe neonatal hypotonia, muscular weakness, feeding problems, and growth failure. To date, only about 10 patients with isolated PREPL deficiency were reported [28,29,30,31,32,33,34]. The affected patients also have facial dysmorphism, motor developmental delay, and urogenital anomalies, as in our patient. The absence of PREPL is associated with reduced filling of the synaptic vesicles with acetylcholine. Pyridostigmine, an acetylcholinesterase inhibitor, improves myasthenic symptoms if the patient is treated early in life, as in our patient [33]. Notably, our patient exhibited multiple malformations, such as hearing loss, supravalvular narrowing of ascending aorta, tethered spinal cord, cryptorchidism, and duplex kidney, which have not been described in CMS22 patients. These additional symptoms could expand the clinical spectrum of this condition. Otherwise, there is the possibility of a complex genetic disease that has not been detected yet in this patient.

AR-DRD, the diagnosis for Patient 2, belongs to a group of hereditary dystonia treatable with dopamine. AR-DRD is caused by a deficiency in tyrosine hydroxylase (TH). Since AR-DRD was first described in 1995 [35], only about 50 cases have been reported [36,37,38,39]. Unlike traditional and common autosomal dominant DRD, known as Segawa syndrome, AR-DRD presents at a younger age, between age 1 and 7 years, and isolated dystonia is uncommon. Infantile Parkinsonism, psychomotor retardation, seizure, hypotonia, ptosis, and autonomic symptoms may develop [39]. TH converts tyrosine to levodopa, and its deficiency disrupts the dopamine biosynthetic pathway. Patient 2 achieved near normal muscle tone and caught-up developmentally with careful levodopa escalation, despite the atypical symptom of infant onset hypotonia without dystonic posture.

EIEE is characterized by progressive diffuse brain dysfunction with recurrent seizures starting during the neonatal or early infantile periods [40]. The incidence is estimated to be 1/50,000 births in the U.K. While seizure with progressive cognitive delay is the core symptom for all EIEE patients, the phenotypes and genotypes are highly variable and the underlying genetic causes are known in only around 50% of the cases [41,42,43]. EIEE 47 is a rare disorder caused by a mutation in the *FGF12* gene and was first reported in 2016 [14]. The mutation is presumed to cause a gain-of-function alteration, which is effectively managed by sodium channel blockers, including phenytoin, as in Patient 3.

EA is characterized by recurrent spells of truncal ataxia and incoordination [44]. Due to the episodic characteristics, EA is difficult to differentiate from other paroxysmal disorders, such as paroxysmal dyskinesia, periodic paralysis, and epilepsy [45]. To date, 8 different subtypes of EA have been defined according to clinical and genetic characteristics and five genes are responsible for EAs. EA2, found in Patient 4, is the most common form of EA, although the prevalence is estimated to be less than 1 in 100,000 with incomplete penetrance [46,47]. The *CACNA1A* gene encodes the CaV2.1 subunit of the P/Q type voltage-gated calcium channel, which is highly expressed in Purkinje cells. The *CACNA1A* mutation results in reduced calcium current and decreased inhibitory effects of Purkinje cells. Acetazolamide responsiveness is a hallmark of the disease and about 50–75% of patients show rapid improvement in episode frequency and severity [47]. Recently, a class I study of 4-aminopyridine (potassium channel blocker) also showed clinical efficacy [48].

Although the number is small, this report of 4 cases is important because the diseases are extremely rare, and their clinical presentations are not specific but effective treatment modalities are available for these children. Ten and 50 cases were reported for the diseases of patients 1 and 2, respectively, and the prevalence of the diseases of patients 3 and 4 were 1–2/10,000. Given the rare incidence, early-onset, non-specific phenotypes, and various genetic etiologies, genome sequencing (WES or WGS) or multi-gene panel testing should be considered for patients showing similar neurological presentations, such as congenital hypotonia, early-onset seizure with progressive cognitive delay, and early onset movement disorders.

WGS is superior to WES in that WGS can detect a deep intronic genetic variant and copy number variation more accurately than WES. However, the overall diagnosis rate of both tests is comparable. In addition, considering the size of the genomic data to be analyzed, 8 GB in WES vs. 150 GB in WGS, WES is less labor-intensive and time-consuming but has a comparable accuracy. For rapid diagnosis with comparable accuracy, WES was chosen in our study. In fact, compared to patient 1 and 2, patient 3 and 4 experienced a long journey until the diagnosis with WES was done.

Importantly, the recent acceleration in the discovery of rare but diverse genetic diseases makes it difficult for physicians to keep updated in the recognition of these genetic diseases. With early application and rapid interpretation of genome studies, physicians can re-phenotype clinical manifestations in patients and overcome the limitations of the usual phenotype-driven strategies. Given the rare incidence, early-onset, non-specific phenotypes, and various genetic etiologies of some diseases, genome sequencing (WES or WGS) or multi-gene panel testing should be considered for patients showing neurological presentations similar to our patients, including congenital hypotonia, early-onset seizure with progressive cognitive delay, and early onset movement disorders. Effective treatment modalities are available for these children.

## 5. Conclusions

In conclusion, accurate genomic diagnosis using WES could help physicians identify the hidden phenotypes and pathophysiologies of extremely rare neurogenetic diseases and, ultimately, provide an effective treatment. WES shifts the patient care paradigm, resulting in improved patient lives.

## Figures and Tables

**Figure 1 jcm-09-03724-f001:**
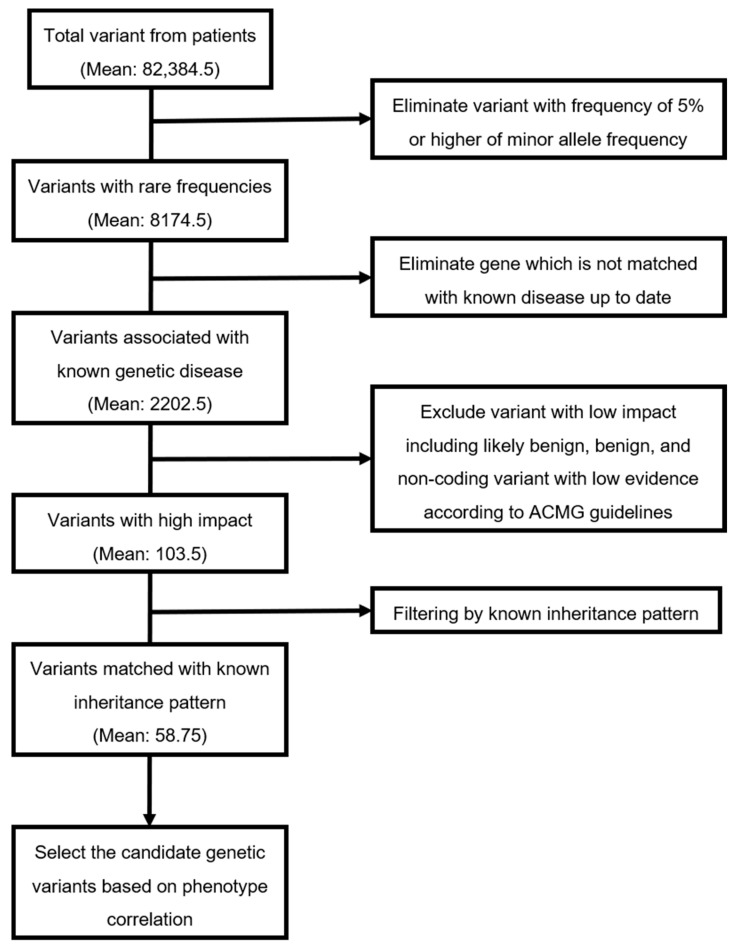
The filtering process to identify the candidate variants.

**Figure 2 jcm-09-03724-f002:**
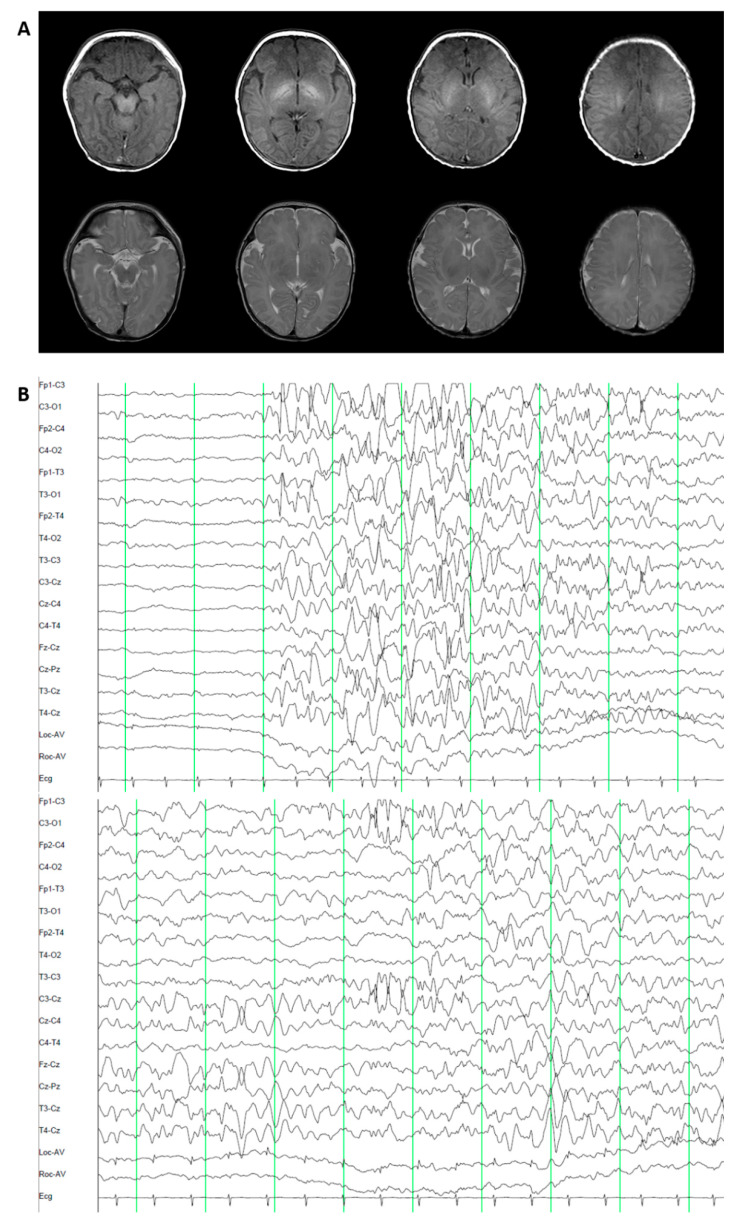
Brain Magnetic Resonance imaging (MRI) and Electroencephalography (EEG) findings of patient 3 at 4 days after birth. (**A**) The T1 (first row) and T2 (second row) images were normal. (**B**) The EEG shows an intermittent suppression-burst background (**top**) and occasional sharp wave discharges from the left occipital and central head areas (**bottom**).

**Figure 3 jcm-09-03724-f003:**
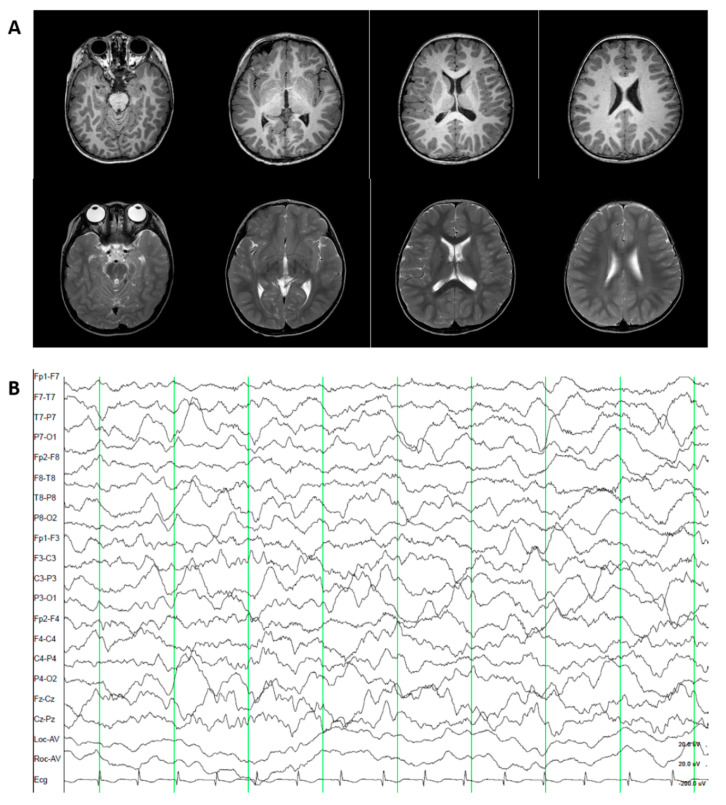
Brain MRI and EEG finding of patient 4 at 2 years. (**A**) The T1 (first row) and T2 (second row) images were normal. (**B**) The EEG was normal.

**Figure 4 jcm-09-03724-f004:**
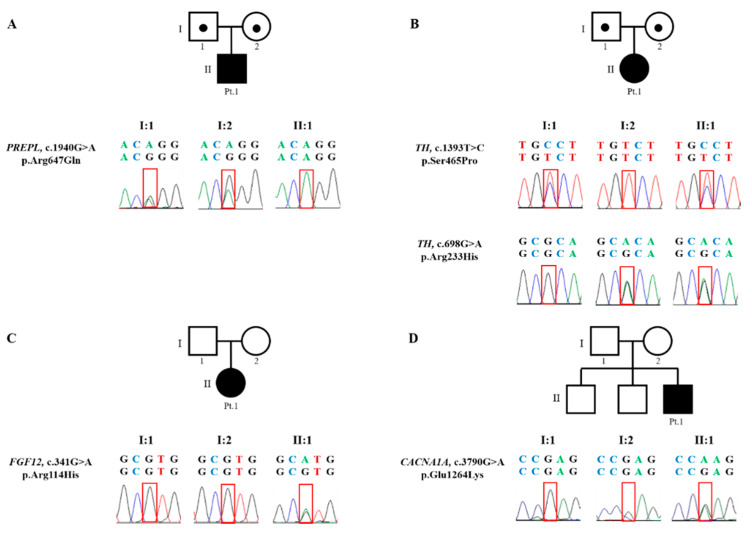
Pedigree and Sanger pherogram of variants in the four patients. (**A**) *PREPL* variants in patient 1 and his parents. (**B**) *TH* variants in patient 2 and her parents. (**C**) *FGF12* variant in patient 3. (**D**) *CACNA1A* variant in patient 4.

**Table 1 jcm-09-03724-t001:** Clinical characteristics of patients with treatable rare neurogenetic diseases.

Characteristics	Patient 1	Patient 2	Patient 3	Patient 4
**Sex**	Male	Female	Female	Male
**Main neurologic** **sign and symptom**	Hypotonia	Global delayed development, hypotonia	Seizure, developmental and epileptic encephalopathy	Episodic ataxia, Global delayed development
**Symptom onset age**	Perinatal period	Infant period	Perinatal period	Infant period
**Final diagnosis**	Congenital myasthenic syndrome 22	Dopa-responsive dystonia	Epileptic encephalopathy 47	Episodic ataxia type II
**Age at diagnosis, year**	2.6	0.8	8.0	4.0
**Family history**	None	None	None	None
**Multisystemic involvement**	Dysmorphic face, hearing disturbance, aortic stenosis, cryptorchidism, duplex kidney	None	None	None
**Involved gene**	Homozygous variant *PREPL* gene (NM_001171603.1) p.Arg647Gln	Heterozygous variants *TH* gene (NM_199292.2) p.Ser465Pro from father p.Arg233His from mother	Heterozygous variants de novo *FGF12* gene (NM_021032.4) p.Arg114His	Heterozygous variants de novo *CACNA1A* gene (NM_001127221.1) p.Glu1264Lys
**gnomAD (total population frequency)**	0.00008501	0.000003992 and 0.0001160	absent	absent
**In silico score**	MutationTaster; Deleterious (1) Polyphen-2:Probably damaging (1) SIFT: Damaging (0)	REVEL: 0.846/0.955	REVEL: 0.548	REVEL: 0.922
**Chromosomal location**	2q21	11p15.5	3q28-q29	19p13.13
**Inheritance pattern**	AR	AR	AD	AD
**Treatment**	Pyridostigmine	Levodopa	Phenytoin	Acetazolamide

SIFT; Sorting intolerant from tolerant; REVEL; Rare exome variant ensemble learner; AR; Autosomal recessive; AD; Autosomal dominant.

**Table 2 jcm-09-03724-t002:** Number of variants of each patient which were carried from whole exome during filtering.

Patients	Patient 1	Patient 2	Patient 3	Patient 4	Average
**Average number of variants from patients**	106,311	107,509	107,600	8118	82,384.5
**Average number of variants with rare frequencies** **(MAF < 5%)**	10,424	10,894	10,463	917	8174.5
**Average number of variants associated with known genetic disease**	2737	2764	2712	597	2202.5
**Average number of variants with high impact**	130	95	94	95	103.5
**Average number of variants matched with known inheritance pattern**	88	46	34	67	58.75

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
