# Peer review of "Clinical Application of Whole Exome Sequencing to Identify Rare but Remediable Neurologic Disorders"

_jcm, 2020, doi:10.3390/jcm9113724_

Round 1

Reviewer 1 Report

I think this is a really well-written paper, and it highlights the importance of carrying out NGS to identify pathogenic variants in order to facilitate rapid diagnosis and effective therapy. There were only a few typos that I found:

1. End of line 162 where you write "On the other hand..." this phrase should be used for where you describe the second mutation. So I would move this phrase to the beginning of the sentence that starts at the end of line 163.

2. In lines 173 and 184, remove "their" from the parts that read "... in the unaffected their parents".

3. Remove "too" from line 273, where you say "Although the number is too small.." as it doesn't make sense to have "too" there. 

Author Response

We really thank again to Journal of clinical medicine and reviewers for the valuable suggestions. We followed your kind advice and tried to revise the paper. The changes are highlighted in yellow shadow lines. Please inspect the new manuscripts and make a good decision.

Reviewer 1

I think this is a really well-written paper, and it highlights the importance of carrying out NGS to identify pathogenic variants in order to facilitate rapid diagnosis and effective therapy. There were only a few typos that I found:

  1. End of line 162 where you write "On the other hand..." this phrase should be used for where you describe the second mutation. So I would move this phrase to the beginning of the sentence that starts at the end of line 163.
  2. In lines 173 and 184, remove "their" from the parts that read "... in the unaffected their parents".
  3. Remove "too" from line 273, where you say "Although the number is too small.." as it doesn't make sense to have "too" there.

Thank you for your good suggestions. We revised the manuscripts and please check the main manuscript.

Reviewer 2 Report

The manuscript by Kim et al reports outcomes from carrying out whole exome sequencing (WES) to diagnose pediatric patients with neurodevelopmental disorders.  The main result is that, out of 376 such patients whose exomes were sequenced, four were diagnosed with disorders where a treatment option was available.  All were subsequently shown to respond positively to treatment.  This work adds to a large body of results that support the diagnostic utility of early use of WES for patients with suspected rare disorders.

In my view there are some issues that need to be addressed prior to publication of this manuscript.  First and foremost, the usual diagnostic success rate for WES is now 35-40%, and this figure is also given in the introduction to this paper (lines 40-41).  So why was only 1.1% of this patient set successfully diagnosed?  Is there something different about this patient population as compared to those used for the studies discussed in references 1-5? Or were other patients diagnosed but not successfully treated?  This needs to be clarified.

Also, in Table 2 the data from Patient 4 are very strange, in that the total number of genetic variants is only around 8% of what is found in the other three patients, yet the number of high-impact variants is roughly the same as the other three.  I don't understand how this is possible if the same variant-calling algorithms were used for all four patients.  Some explanation of this variance needs to be provided.

Minor comments:

On lines 58-59 it is surely an overstatement to state that 'all exons of all human genes' were captured in the selection.  Comparison of WGS and WES data generally shows that a small minority of exons are underrepresented or absent in WES data for a variety of reasons.

On line 184 the word 'their' should be deleted after 'unaffected'.

Author Response

To Reviewers

We really thank again to Journal of clinical medicine and reviewers for the valuable suggestions. We followed your kind advice and tried to revise the paper. The changes are highlighted in yellow shadow lines. Please inspect the new manuscripts and make a good decision.

Review 2

The manuscript by Kim et al reports outcomes from carrying out whole exome sequencing (WES) to diagnose pediatric patients with neurodevelopmental disorders. The main result is that, out of 376 such patients whose exomes were sequenced, four were diagnosed with disorders where a treatment option was available. All were subsequently shown to respond positively to treatment. This work adds to a large body of results that support the diagnostic utility of early use of WES for patients with suspected rare disorders.

  1. In my view there are some issues that need to be addressed prior to publication of this manuscript. First and foremost, the usual diagnostic success rate for WES is now 35-40%, and this figure is also given in the introduction to this paper (lines 40-41). So why was only 1.1% of this patient set successfully diagnosed? Is there something different about this patient population as compared to those used for the studies discussed in references 1-5? Or were other patients diagnosed but not successfully treated? This needs to be clarified.

Thank you for your good questions. The diagnostic yield of WES for these cohort were 42.7% (Previously published report in Clinical genetics, 2020, “Diagnostic yield and clinical utility of whole exome sequencing using an automated variant prioritization system, EVIDENCE”) and only 4 patients (1.1%) have ‘treatable’ rare neurologic disease. We focused on the patient with remediable neurogenetic disorders and did not showed the data of genetic diagnosis in whole patients.

  1. Also, in Table 2 the data from Patient 4 are very strange, in that the total number of genetic variants is only around 8% of what is found in the other three patients, yet the number of high-impact variants is roughly the same as the other three. I don't understand how this is possible if the same variant-calling algorithms were used for all four patients. Some explanation of this variance needs to be provided.

Thank you for your point out. The patients 4 performed the panel test which identify the approximately 62,000 exomes in 4,813 genes instead of WES. The filtering process (Figure 1) to find variants was same as other patients. We also revised the method and result of manuscript.

Minor comments:

  1. On lines 58-59 it is surely an overstatement to state that 'all exons of all human genes' were captured in the selection. Comparison of WGS and WES data generally shows that a small minority of exons are underrepresented or absent in WES data for a variety of reasons.

Thank you for your good advice. We revised the sentence to “Majority of exons in all human genes”. Please check the main manuscript.

  1. On line 184 the word 'their' should be deleted after 'unaffected'.

Thank you for your good suggestions. We revised the manuscripts and please check the main manuscript.

Reviewer 3 Report

Kim MJ et al investigated rare mutations in 376 children with neurodevelopment 

al symptoms using exome sequencing. They identified 4 patients with with treatable neurologic disorders. The mutations are found in PREPL, TH, FGF12 and CACNA1A. This is a well-written nice report. 

I have a few minor comments/questions/suggestions to the authors:

  1. Why did you choose minor allele frequency of 5% to filter out common variants? Isn't it too high for rare diseases?
  2. Did you you use any in silicon gene panel for neurodevelopment diseases? If, yes, please provide the list of genes as a supplementary material.
  3. Please provide minor allele frequencies from gnomAD, CADD scores, other in silicons predictions, inheritance pattern and chromosomal location (Hg19) in a table.
  4. Please give more detail about the variant processing and annotation in the material methods (the softwares, tools that were used and their version numbers).
  5. What is the read depth and coverage for exome sequencing?
  6. In line 88, you wrote "... of 62,000 OMIM genes..." Is this correct? Please check the number of OMIM genes again.

Author Response

To Reviewers

We really thank again to Journal of clinical medicine and reviewers for the valuable suggestions. We followed your kind advice and tried to revise the paper. The changes are highlighted in yellow shadow lines. Please inspect the new manuscripts and make a good decision.

Review 3

Kim MJ et al investigated rare mutations in 376 children with neurodevelopmental symptoms using exome sequencing. They identified 4 patients with with treatable neurologic disorders. The mutations are found in PREPL, TH, FGF12 and CACNA1A. This is a well-written nice report.

I have a few minor comments/questions/suggestions to the authors:

  1. Why did you choose minor allele frequency of 5% to filter out common variants? Isn't it too high for rare diseases? 

Thank you for your point-out. The filtering algorithm is not only for detecting the rare diseases, but also for identifying the other genetic disorders. We excluded the variants with a frequency of 5% or higher in the minor allele frequency according to the rule BA1 of the ACMG guidelines.

  1. Did you you use any in silicon gene panel for neurodevelopment diseases? If, yes, please provide the list of genes as a supplementary material.

Thank you for your good questions. Diagnostic exome panel test was performed on patient 4 only. We added the data of panel test in method and results of manuscripts and provide the list of genes as a supplementary material 1.

  1. Please provide minor allele frequencies from gnomAD, CADD scores, other in silicons predictions, inheritance pattern and chromosomal location (Hg19) in a table.

Thank you for your good suggestions. We revised the table and please check the main manuscript.

  1. Please give more detail about the variant processing and annotation in the material methods (the softwares, tools that were used and their version numbers).

Thank you for your good advice. We used the steamlined, automated variant prioritization software system, termed EVIDENCE (3bilion Inc., Seoul, South Korea) used to identify the candidate variants. The software program analyses over 100,000 variants, according to ACMG guidelines, prioritize the variants based on each phenotype of each patient and interpret these variants accurately and consistently. We also add the detail method in manuscript.

  1. What is the read depth and coverage for exome sequencing?

Thank you for your good questions. We provide the read depth and coverage for exome sequencing in supplementary material 2.

  1. In line 88, you wrote "... of 62,000 OMIM genes..." Is this correct? Please check the number of OMIM genes again.

Thank you for your good suggestions. We recheck and revised the sentence to “4,813 OMIM genes”.

Round 2

Reviewer 2 Report

My previous comments have been satisfactorily addressed.  Only some English errors in the newly added text need to be corrected.

line 67, streamlined is misspelled.

line 68, replace 'are' with 'is'

line 69, replace 'prioritize' with 'prioritizes'

line 70, replace 'interpret' with 'interprets'

Author Response

Thank you for your good advice. Your valuable suggestions guided us to the much better report. We rechecked the errors. Please see the revised manuscript.